# Organosilicon-Based Plasma Nanocoating on Crust Leather for Water-Repellent Footwear

**DOI:** 10.3390/ma15207255

**Published:** 2022-10-17

**Authors:** Carlos Ruzafa-Silvestre, Blanca Juan-Fernández, María Pilar Carbonell-Blasco, Elena Bañón-Gil, Elena Orgilés-Calpena, Francisca Arán-Ais

**Affiliations:** Footwear Technology Centre, Campo Alto Campground, 03600 Alicante, Spain

**Keywords:** leather, hydrophobic, coatings, finishing, hexamethyldisiloxane, footwear, sustainable

## Abstract

In this study, functional nanocoatings for water-repellent footwear leather materials were investigated by chemical plasma polymerisation by implanting and depositing the organosilicon compound hexamethyldisiloxane (HMDSO) using a low-pressure plasma system. To this end, the effect of monomers on leather plasma deposition time was evaluated and both the resulting plasma polymers and the deposited leather samples were characterised using different experimental techniques, such as: Fourier transform infrared spectroscopy (FTIR), X-ray photoelectron spectroscopy (XPS) and scanning electron microscopy (SEM). In addition, leather samples were tested by standard tests for color change, water resistance, surface wetting resistance and dynamic water contact angle (DWCA). The resulting polysiloxane polymers exhibited hydrophobic properties on leather. Furthermore, these chemical surface modifications created on the substrate can produce water repellent effects without altering the visual leather appearance and physical properties. Both plasma coating treatments and nanocoatings with developed water-repellency properties can be considered as a more sustainable, automated and less polluting alternative to chemical conventional processing that can be introduced into product-finishing processes in the footwear industry.

## 1. Introduction

The hydrophobicity of materials is a key performance in the footwear finishing process, where it is ensured that the surface of the material is optimally treated in order to functionalise it in a suitable way to meet the water repellence requirements for footwear applications [1,2].

One of the most commonly used substances in the finishing process, and directly involved in the wettability of the material, are coatings, whose purpose is to improve certain properties or qualities of the surface of materials. The choice of base materials determines the coating properties, such as hydrophobicity (aforementioned), hydrophilicity, scratch protection, diffusion barrier, slip/anti-stick coatings, antimicrobial, adhesion promoter, and water/water vapour barrier, among others [3,4,5].

Chemicals based on halogen organic compounds and fluorocarbon compounds, both with hydrophobic properties and adequate durability, are currently used, especially for the formulation of durable water repellent (DWR) coatings, which have no wettability providing water resistance to different surface materials [6,7,8,9]. These chemicals, which are considered hazardous, are limited and/or restricted nowadays by current European regulations [10].

In terms of their application in footwear materials, they are mainly applied on upper materials, whether leather [11] or textile [12], to provide them with water-repellent properties or repellence to aqueous solutions. This is due to the fact that these materials have a high wettability thanks to their composition or nature. In addition, the conventional processes used for their application are water-based, involve high energy consumption, large volumes of water and a high range of hazardous chemicals, as well as the emission of volatile organic compounds (VOCs) [13,14]. Therefore, the industry requires new, innovative and more resource-efficient production processes that minimise the use of hazardous chemicals, as well as the volume of waste, effluents and emissions generated.

One of the environmentally sustainable and efficient alternatives for coating materials, which is already applied in other industrial sectors, is plasma technology. This technology is based on the premise that a continuous application of energy to a gas produces an electric charge on its particles, which causes them to lose their electromagnetic equilibrium and the particles to ionise, forming atoms that are not electrically neutral and are then called plasma [15].

Currently, the most widely used industrial plasma technologies are both low-pressure plasma (LPP) and atmospheric pressure plasma jet (APPJ). Among the treatments offered by plasma technologies, there is the coating treatment of materials based on chemical plasma polymerisation [16]. Through this process, a gaseous or liquid monomer/chemical precursor can be polymerised, coating and fully functionalising the material by forming an ultra-thin, structured layer of the polymer on the surface of the material, providing a specific surface property on it [17,18,19].

Plasma-coating technology is considered an environmentally sustainable alternative as it helps to reduce the environmental impact of the waterproofing process and the final product, and to contribute to the decarbonisation of footwear manufacturing. The significant environmental benefits provided by the plasma-coating process compared to the conventional wet chemical process are as follows according to the literature that has investigated it [14,20]: coating chemical consumption is reduced by 80% because the coating applied to reach that functionality is much thinner; water consumption is reduced by 100% and energy consumption by 50%, due to the fact that the plasma process is totally dry and does not require any curing steps; and finally, no cross-linking agents, chlorides, formaldehyde, etc., found in wet chemical coatings of toxic products are required.

Regarding the composition of the molecular structure and the functional groups exhibiting hydrophobic properties, different precursor chemicals can be used to make different plasma polymerised coatings. All precursor compounds that have hydrophobic qualities are classified in four groups: (a) halocarbon compounds (chlorocarbon, fluorocarbon) that form polymers with an organic molecular structure -C-C- and hydrophobic chemical groups CClx, CClFx and CFx; (b) hydrocarbon compounds leading to coatings with the organic structure -C-C- and hydrophobic alkyl (CHx) chemical groups; (c) fluorosilane compounds with (inorganic) siloxane bonding -Si-O-Si- and hydrophobic CFx side groups; and (d) organosilicon compounds for coatings with hydrophobic bonding -Si-O-Si- and hydrophobic CHx side groups [21].

Although the compounds of the first three subcategories have a high hydrophobic power on surfaces and perform adequately, generating high-quality and durable coatings, given their high toxicity and danger to human health and the environment [22], it is necessary to investigate alternative compounds that also possess hydrophobic properties, such as those comprising the fourth subcategory—organosilicon compounds [23]—but which are safer with a lower environmental footprint.

One of the most studied organosilicon compounds as a hydrophobic plasma coating is hexamethyldisiloxane (HMDSO) because, in its plasma-excited state, it fragments and can generate silicon radicals and atoms of silicon, hydrogen, carbon and oxygen [24,25]. These are deposited on the surface of the treated material, leading to a strong interaction between them. As a result, a small layer of the polymerised HMDSO that coats the entire surface of the material is applied. From the 1970s to date, relevant works by different authors have investigated the line of research that concerns this study: the plasma polymerisation process, how the working parameters influence the final coating and its properties, the application of this process on different materials and its possible copolymerisation with other chemical precursors [26,27,28,29]. However, few authors have studied the plasma polymerisation process of different organosilicon compounds, such as HMDSO on leather and synthetic materials [30], for the development of hydrophobic coatings [31,32,33,34]. Previous work by the authors showed preliminary evidence of the use and optimisation of plasma technology [35,36,37] with enhanced hydrophobicity at high water contact angles in leather for footwear applications [38,39].

Therefore, as a continuation of a previous work, in this study, the LPP-operated polymerisation with HMDSO has been used as an organosilicon component to deposit coatings with various new and hydrophobic chemical functional groups from this precursor on leather materials for footwear applications. To understand the influence of plasma polymerisation of a precursor on the formation of the different functional groups, a comparison with uncoated samples was made. Furthermore, the formation of silicon-containing groups and how the siloxane network structure depends on the deposition time of the precursor chemical was studied. A better understanding of the mechanism of the single-step plasma polymer formation process—chemical plasma polymerisation—has allowed a reduction of the coating process time for the intended leathers with a promising result for water-repellent leather performance, which can be of great interest for the leather, footwear or even textile and clothing (TCLF) industries to reduce their environmental footprint contributing also to their decarbonisation [40].

## 2. Materials and Methods

### 2.1. Materials

In this work, hexamethylsidiloxane (HMDSO), ((CH_3_)_3_-Si-O-Si-(CH_3_)_3_, 98% purity) provided by Merck Life Science S.L.U. (Madrid, Spain) was used as a hydrophobic liquid chemical precursor as received. A crust leather of bovine origin with chrome tanning and vegetable retanning, without surface finishing and dyed in brown, supplied by the Spanish company Pies Cuadrados Leather S.L. (Aspe, Spain), was used for the plasma polymerisation processes as a representative upper material for footwear applications.

### 2.2. Plasma Polymerisation Coating Process

Plasma-polymerised HMDSO films were prepared on leather samples in nano low-pressure plasma equipment (Diener Electronic Vertiebs GmbH, Ebhausen, Germany), of modular configuration, with a 24 L volume chamber, in stainless steel and with a tray for sample support. The plasma reactor has a 13.56 MHz radio frequency (RF) generator with a maximum power of 300 W. It is also equipped with two gas supply channels and a micro-dosing pump. It performs all types of plasma surface treatment processes: cleaning, activation, coating and etching at a laboratory scale. The thermostatised treatment chamber allows the temperature to be raised during the plasma process, which increases the effectiveness of the HMDSO film deposition process. The parameters of the coating treatments were set by varying the plasma exposure time between 180 to 720 s, and setting the plasma input power at 100 W. The nomenclature of the samples with the different working parameters is shown in Table 1.

The plasma coating comprises a single-step chemical polymerisation process, as can be seen in Figure 1. Leather samples with a size of 148 × 210 mm were introduced into the chamber of the low-pressure plasma equipment. Immediately, the pressure in the chamber was reduced to the required vacuum level. The coating monomer, HMDSO, was then introduced through the mass flow controller and allowed to stabilise until a certain pressure was reached. In this way, the monomer molecules were distributed evenly throughout the chamber. After the gas supply time had expired, the generator was switched on. The molecules dispersed in the chamber were ionised, generating HMDSO plasma that spread throughout the chamber, modifying the exposed surface of the samples. The plasma was released for the programmed time so that the coating was deposited on the leather sample. Finally, the chamber was pressurised by introducing vent air until atmospheric pressure reached into the chamber. The samples were then extracted from inside the chamber [39,41,42].

### 2.3. Characterisation Techniques

#### 2.3.1. Fourier Transform Infrared Spectroscopy (FTIR)

The chemical modifications of the surface of the plasma-coated leather, as well as the identification of organosilicon hydrophobic functional groups, were characterised by using a Varian 660-IR infrared spectrophotometer (VARIAN Australia PTY LTD; Mulgrave, Australia). Attenuated total reflectance (ATR) mode with 16 scans at a resolution of 4 cm^−1^ was used as the FTIR sampling technique. This ATR accessory works by measuring changes in the infrared beam when the beam comes into contact with the sample.

#### 2.3.2. X-ray Photoelectron Spectroscopy (XPS)

The surface chemical content of the siloxane polymer film obtained by plasma polymerisation was examined using an X-Ray Photoelectron Spectrophotometer (XPS, K-ALPHA, Thermo Scientific, Waltham, MA, USA). To determine the chemical characteristics of the ultrathin films, XPS was chosen as the best technique. XPS data were collected at 3 mA and 12 kV using K-ALPHA (Al-K) radiation (1486.6 eV), monochromatised by a double crystal monochromator, and a focused X-ray spot (elliptical in shape with an axis length greater than 400 m) was obtained. The alpha hemispherical analyser worked in continuous energy mode, scanning through the 200 eV energy band to measure the entire energy band and then using 50 eV in a narrow scan to probe individual elements. The XPS data were processed with the Avantage software, and the smart background function was used to approximate the experimental background and calculate the elemental composition of the surface based on the peak area removed from the background. The system’s flood gun, which produces low-energy electrons and low-energy argon ions from a single source, was used to perform charge compensation. The Technical Research Services of the University of Alicante (SSTTI) performed this analysis (UA).

#### 2.3.3. Scanning Electron Microscopy (SEM)

Surface modifications and morphological analysis of the coated and uncoated leather samples were carried out with a Jeol model IT500HR/LA high-resolution scanning electron microscope provided with EDS analysis. It is equipped with a field emission gun which provides high resolution (1.5 nm at 30 kV, 4.0 nm at 1 kV) and can work in a voltage range from 0.5 to 30 kV. The microscope is also equipped with a Raith Elphy Quantum electron beam nanolithography (EBL) system. This analysis was carried out by the Technical Research Services (SSTTI) of the University of Alicante (UA).

#### 2.3.4. Colour Difference

The colour difference of plasma-treated leathers was measured with a CM600d spectrophotometer in accordance with ISO 22700: 2019 [43]. This portable spectrophotometer is designed to assess the colour and appearance of samples of various sizes, including the surfaces of flat, shaped or curved items. It features a fixed 8 mm aperture and two measurement modes to suit the surface conditions of each sample: specular reflectance included (SPINC) and specular reflectance excluded (SPEX), the latter being used for the measurement as it takes into account the polish of the sample surface. The measurements were carried out at three different points in the centre of the sample.

#### 2.3.5. Water Repellency Properties

The aqueous liquid repellency test is described in ISO 23232: 2009 [44]. Eight drops of eight solutions of different water and isopropyl proportions are placed in a staggered manner, i.e., the first solution with 100% water and the last one with 60% isopropyl, on the surface of the substrate. After this, the adsorption, wicking and contact angle on the surface of the studied material is observed. Thus, the aqueous repellency grade is the highest numbered test liquid which is not absorbed by the substrate surface.

#### 2.3.6. Surface Wetting Resistance

The test for the water repellency of leather is set out in ISO 17231: 2017 [45]. This test describes a method for determining the resistance of leather to surface wetting. It is applicable to all leathers intended for apparel manufacturing. The spray index is the measure of the resistance of a leather surface to wetting. To determine the spray index, the appearance of the test piece is compared with descriptive and photographic patterns.

#### 2.3.7. Dynamic Water Contact Angles (DWCA)

Dynamic contact angles (DCA) [46,47], including the advancing (ϴ_A_) and receding (ϴ_R_) contact angles by which the hysteresis contact angle (ϴ_H_) is obtained, of uncoated and plasma-coated leather samples were measured with an Attention Theta Flow optical tensiometer (Biolin Scientific Oy, Espoo, Finland). These were performed using three drops of bi-distilled water on a rectangular sample as measuring liquid by the needle method, which is specifically used to measure dynamic contact angles on superhydrophobic surfaces. Then, the values were calculated by the polynomial method according to ISO 19403-6: 2017 [48].

## 3. Results and Discussion

This section will provide a precise description of the experimental results, their interpretation, as well as the experimental conclusions that can be drawn.

### 3.1. Chemical Properties Characterisation of the HMDSO Plasma Polymerised Coated Leather Samples

Figure 2 shows the FTIR spectra of the uncoated and plasma-coated leather samples with the main bands and functional groups of the leather and organosilicon coating. On the one hand, the CC-O as control leather sample showed sharp absorption peaks located at 1652 cm^−1^ associated with the C=O amide in the peptide band (Amide I). The 1538 and 1751 cm^−1^ peaks represented the N-H of Amide II and C=O stretching due to the ester fatty acids, respectively. The -CH *stretching vibration mode* (*st*) was related to the bands around 2915 cm^−1^ and was quite stable on the leather surface. Furthermore, amide A band appeared around 3300 cm^−1^ due to the stretching vibration of -NH groups and the conformation of the backbone, which was very sensitive to the strength of the hydrogen bonds [49].

On the other hand, infrared spectra corresponding to the crust leather samples coated with HMDSO at different exposure times are also included in Figure 2. In these spectra, the typical bands of the mentioned leather as well as the characteristic bands of the organosilicon monomer deposited on its surfaces were observed [50]. For instance, the bands at 800 and 840 cm^−1^ correspond to the Si-C *st* and Si-(CH_3_) out-of-plane bending vibration (*γ*) and start to be observed already in sample CC-3, their presence becomes more noticeable in sample CC-6, and in sample CC-12, they are easily distinguishable, forming those two peaks. The 1040 cm^−1^ band corresponds to the Si-O-Si bonds which quantitatively increase as the processing time increases, causing the area of the peak to become progressively larger, after applying the HMDSO plasma polymerisation. Additionally, the 1257 cm^−1^ peak corresponding to the Si-(CH_3_) bending symmetric vibration (*δ_sy_*) is clearly observed in CC-12 sample, with the longest exposure time, while in the other samples it hardly appears at all. FTIR analysis confirms how the polysiloxane-based layer has been formed on the leather surface of the leather after increasing the exposure time of the plasma-coating process, which has enhanced the deposition and homogenisation of the silane-based film [51].

An XPS study was necessary to comprehensively analyse the chemical modifications on the outermost surface of uncoated and plasma-coated leather samples. Figure 3 shows the results of the XPS-survey, in which it can be observed that the CC-O sample was mainly composed of oxygen (O 1s), nitrogen (N 1s) and carbon (C 1s), whose peaks were positioned at about 532, 400 and 285 eV, respectively. The plasma-coated samples showed changes in the intensity of the O, N and C peaks, as well as the appearance of new clearly visible Si 2s and Si 2p bands with binding energies at 154 and 103 eV, respectively. These chemical modifications, with respect to the uncoated sample, were due to the silicon coating deposited because of the dissociation of the HMDSO monomer by plasma irradiation. This occurs mainly due to the formation of two free radicals, (CH_3_)_3_-Si-O and Si-(CH_3_)_3_, precursors of the film growth [37,50,51].

The elemental composition of the HMDSO-based coatings on the leather samples were determined by using XPS spectroscopy. The results are included in Table 2. After deposition of the monomer on the leather samples, a decrease in the percentages of carbon and oxygen was observed, much more marked in nitrogen content. However, the silicon content increased significantly in all samples by more than 6% after increasing the deposition time from 180 to 720 s. These results confirmed that a silica-based coating is deposited on the leather surface by the oxygen depletion and total coating of the nitrogen groups of the leather provided by the amides that compose it [24,30,31].

To obtain information on the chemical species formed in HMDSO coatings, C 1s and Si 2p peaks were deconvoluted, as shown in Figure 4, and their atomic concentrations are reported in Table 3. On the one hand, it was observed that samples CC-0 and CC-3 showed a similar XPS spectrum, and totally different from samples CC-6 and CC-12. In the case of samples CC-0 and CC-3, the presence of carbon chemical species with different oxidation states, such as C-C/C-H, C-N/C-O, C=O, O=C-O, and COOH/COOR, located at 284.6, 286.0, 286.9, 288.7 and 291.5 eV, were observed, being more pronounced in sample CC-3 due to a possible partial oxidation of the carbon-containing fragments of HMDSO. It is also possible that a new functional group of the HMDSO structure, C-Si at 286.0 eV, appeared in this sample [29]. In the case of samples CC-6 and CC-12, the spectra are similar with a large peak of C-C/C-H bonds on the surface and with a small peak attributed to C-Si at 286.0 eV, due to the introduction of the methyl groups present in the HMDSO monomer after its atomic fragmentation, indicating a decrease of the surface oxidation and removing the polar character of the surface. This seems to confirm that the plasma exposure time for an effective deposition and formation of the HMDSO polymeric film could be longer than 180 s [24,52,53,54,55].

On the other hand, the intensity of the Si 2p peak of the uncoated sample is so low that it can be considered negligible. However, in all plasma polymerised coated samples, the appearance of this silicon peak was clearly visible, which could be deconvoluted showing the following organosilicon chemical species: silicon oxides (SiO_x_) at 101.30 eV and polydimethylsiloxane (SiO_2_(CH_3_)_3_) at 103.20 eV.

In Figure 4 and Table 3, it is observed that the atomic content of SiOx increased with increasing deposition time in the coated samples. In addition, the plasma coating generates the appearance of a characteristic peak of the SiO_2_(CH_3_)_3_ species, which becomes more intense as the deposition time increases. This means that polymerisation of the free radicals generated during the dissociation of HMDSO leads to the deposition of a polysiloxane film, which is enhanced by the exposure time of the leather samples to the plasma coating.

All these surface chemical modifications analysed with XPS justify the formation and suitable deposition of an ultra-thin layer with an organosilicon SiO_x_C_y_H_z_ structure on the leather surface and corroborate the FTIR results previously described [56].

### 3.2. Physical Properties Characterisation of the HMDSO Plasma Polymerised Coated Leather Samples

Figure 5 shows the SEM images taken of the HMDSO plasma polymerised coated and uncoated leather samples. It was observed that in (b–d) images, the surface is slightly smoother and the leather fibres appear more flattened and not as loose as in sample (a). In addition, it should be noted that the leather samples have characteristic pores in different sizes, as can be seen in the images. The pores of the three samples were not affected after the treatment, as they remained the same size, which is good for a proper perspiration of the leather as upper material, indicating that it is not affected by the HMDSO plasma polymerised. The observed morphological changes suggested that a very ultra-thin layer of the deposited organosilicon coating had been created, thus covering only the leather fibres and making them more compact [57,58,59].

In addition, possible colour changes of leather surfaces of the plasma-coated and uncoated samples have been evaluated by the colour space L* (lightness), a* (red/green), b* (yellow/blue), also referred to as CIELAB according to the standard ISO 22700: 2019, as shown in Figure 6. The obtained results showed no significant difference in colour between the coated samples respect to the untreated sample. Furthermore, the total colour difference (ΔE*) is measured according to the requirement set by INESCOP’s upper materials lab at ≤2.5, and samples CC-3, CC-6 and CC-12 showed lower values: 0.43, 0.60 and 1.06, respectively. It should be noted that ΔE* values seemed to increase slightly but not significantly, which may be due to the increase in the exposure time of the HMDSO plasma polymerisation treatment, which also increases the concentration and quantity of monomer inside the plasma system chamber and on the surface of the leather samples. Therefore, it can be concluded that the plasma polymerisation process of HMDSO on the crust leather does not affect the colour of the samples, keeping the pigment and the appearance of the leather after treatment as it was [30,60].

### 3.3. Hydrophobicity and Water Repellence Assessment

The Aqueous Isopropyl Solution Repellency Grade is determined for all the leather samples according to the ISO 23232. Grade A indicates the drop is clear on the sample, while for grade B the contour begins to darken. In case C, there is apparent wicking and/or complete wetting, and, in case D, the sample is completely wet. When the results obtained were analysed (see Table 4), it was observed that the control sample repelled pure water and remained completely wet in the 30% isopropyl solution. However, sample CC-3 showed repellency grade A for the first three solutions, remaining wet for the 40% solution, going from grade B to D. For sample CC-6, the behavior observed for grade A was similar to CC-3, while grade C is obtained, remaining completely wet for the 60% isopropyl solution. Finally, sample CC-12 obtained a higher-grade A, being completely wet with the 70% isopropyl solution. Therefore, the deposition time of the HMDSO plasma on the samples is proportional to the degree of repellency, increasing from samples CC-3 to CC-12. This result is in agreement with those obtained in FTIR and XPS, which showed the formation of a noticeable siloxane layer with the corresponding hydrophobic functional groups, as the deposition time was increased [61].

After spraying the leather with water (Figure 7), it was observed that the percentage of absorption of the control sample was the highest, producing the wetting of the entire surface with a spray index equal to one according to the photographic patterns of the standardised test. In the case of the already treated samples, the percentage of absorbed water was much lower, corresponding to a degree of wettability equal to four, for which wetting did not occur, but small drops remained attached to the wetted surface. It can be concluded that there is no clear difference between the results obtained for the treated samples, as the same spray rate is obtained although the percentage of water absorption is slightly different but not significant. This determines that all HMDSO plasma treatments achieved a suitable water repellency due to the silane-based layer formed on the leather surface [62].

The results obtained from the dynamic contact angle measurements with distilled water, after applying the HMDSO plasma coating and without it, are provided in Table 5, together with the droplets tested in Figure 8. The uncoated sample surface CC-0 is chemically hydrophilic and heterogeneous, causing the advancing contact angle to increase rapidly to a high value even if the surface is only slightly covered with hydrophobic inclusions. However, after plasma application, it was observed that the advancing contact angles increased in all the samples with respect to the untreated sample, from 114° to 158°, the latter being the highest obtained for sample CC-12. In addition, the same behaviour was observed for the receding contact angle, although the increase in the value is much higher, going from 67° for the untreated sample to 158° for CC-12 sample. Finally, it was observed that the hysteresis contact angle, which is the difference between the advancing and receding angle, decreased as the application time of the HMDSO plasma polymerisation process on the leather sample increases, as can be seen in Figure 9. This modification was due to the surfaces with a lower value of hysteresis angle suggesting a higher uniformity in the deposited coating, and thus higher hydrophobic properties, acquired due to the organosilicon functional groups formed [63]. Therefore, sample CC-12 could be considered as superhydrophobic, also supported by the fact that the obtained receding contact angle is higher than 150° [19,39,64,65].

## 4. Conclusions

This study has proposed the use of low-pressure plasma technology to alter the surface properties of leather to make it hydrophobic, thus increasing the usage of leather. A one-step plasma polymerisation coating process with an organosilicon precursor (HMDSO) on leather for footwear application was investigated using an LPP system and optimising the working parameters. It was observed that in LPP polymerisation, the deposition time was a key parameter for the proper formation, cross-linking and coating of the ultra-thin layer created by the siloxane on the leather surface. The suitable plasma hydrophobisation process of the leather material is supported by the following physiochemical evidence concluded from the results obtained in this research:–Organosilane and methyl-based chemical functional groups were introduced into the chemical structure of the leather surface due to the ionisation of the HMDSO monomer, which mainly offered and provided the materials with water repellent properties.–The physical characterization by different standardised tests revealed that the properties of the treated leather samples, such as colour or texture, were not modified. This is an advantage over conventional methods of functionalising and finishing leather, as the application of wet-end finishes sometimes modifies the perceived colour shades of the material.–Plasma coatings significantly increased the hydrophobic properties of all coated leather surfaces, but this was especially pronounced for specific parameter sets, as in sample CC-12, which showed the highest water resistance and water repellency.–The deposition time of the plasma coating process influenced the final performance obtained. Since a nano-sized film of polymer-functional plasma must be generated to coat the leather surface, long times (720 s) were more favourable for this purpose than short times (180 s).–Suitable performance of the chemical precursor HMDSO for use in LPP technology as a polymeric coating to impart water repellent and resistant properties to leather materials for footwear applications.

From the point of view of the finishing process in the footwear industry, the results obtained are promising and demonstrate that low-pressure plasma technology can be used for the hydrophobisation of the leather used, and thus improving the use of the final product by the consumer, providing them with high added value. In addition, this water-repellent coating treatment also contributes to reducing both the process and the product environmental footprint, thus helping the decarbonisation of the footwear manufacturing sector. Compared to traditional water-repellent processes, which are wet and chemical processes, plasma coating has significant environmental benefits, such as reducing the use of hazardous chemicals and preserving water and energy, thus being more resource-efficient. Finally, it should be noted that the precursor chemical used in the leather coating has a halogen-free chemistry, making plasma technology a potential innovative solution in the finishing process for the leather and footwear industry [40,66].

## Figures and Tables

**Figure 1 materials-15-07255-f001:**
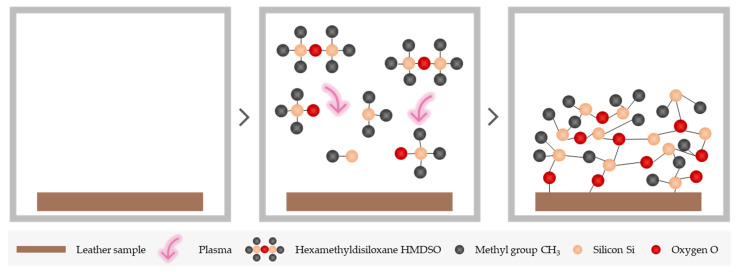
HMDSO single-step plasma polymerisation process.

**Figure 2 materials-15-07255-f002:**
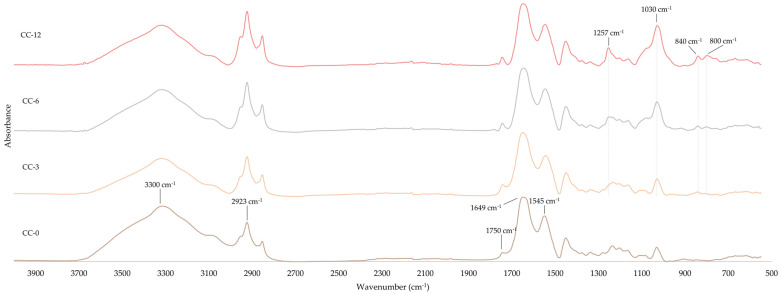
FTIR spectra of the untreated leather sample and leather samples coated with plasma-polymerised HMDSO.

**Figure 3 materials-15-07255-f003:**
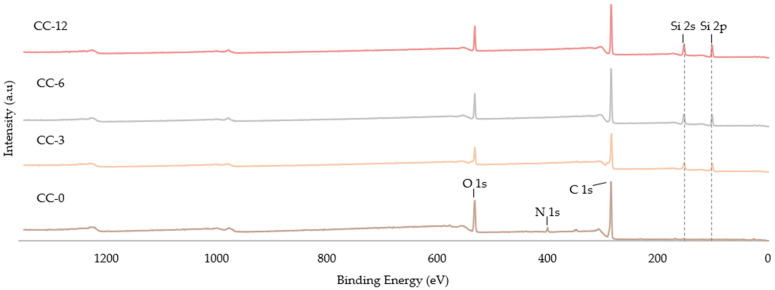
XPS survey of untreated leather and plasma-coated leather samples with HMDSO.

**Figure 4 materials-15-07255-f004:**
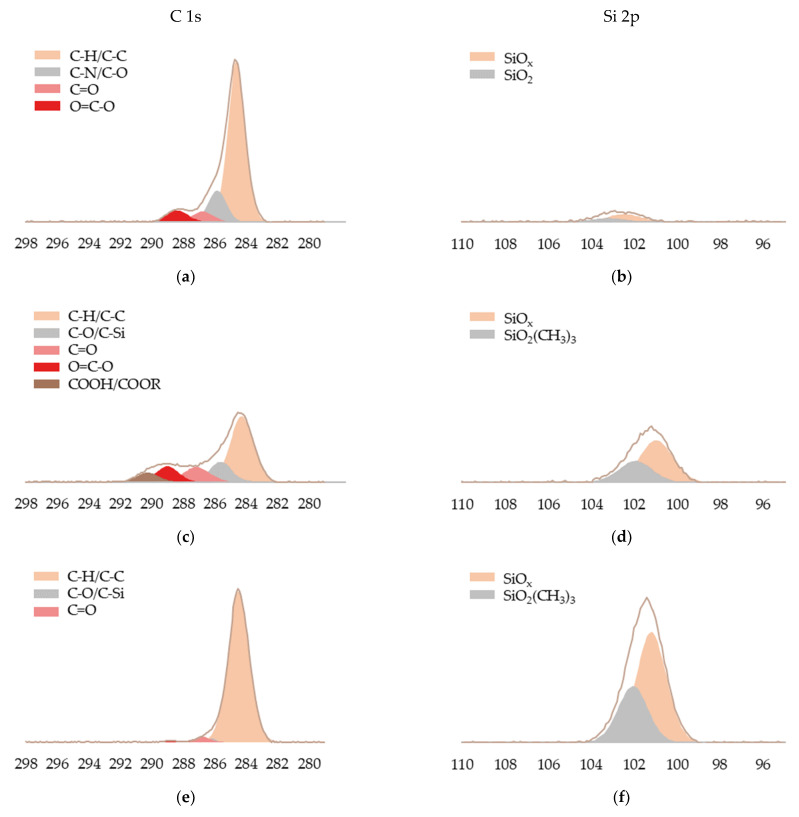
Deconvoluted C 1s and Si 2p peaks of HMDSO films deposited on leather samples. (**a**) C 1s of CC-0; (**b**) Si 2p of CC-0; (**c**) C 1s of CC-3; (**d**) Si 2p of CC-3; (**e**) C 1s of CC-6; (**f**) Si 2p of CC-6; (**g**) C 1s of CC-12; (**h**) Si 2p of CC-12.

**Figure 5 materials-15-07255-f005:**
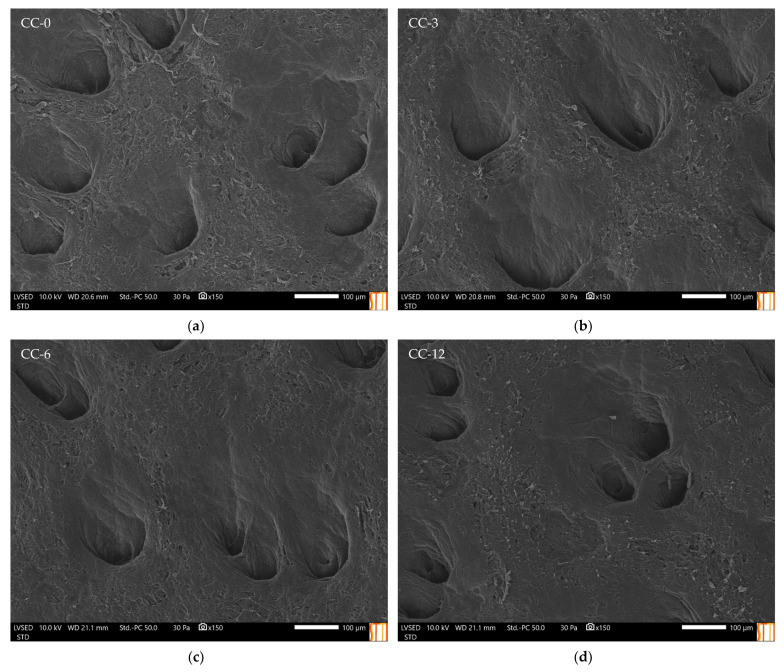
SEM micrographs of plasma-coated and untreated leather samples at 150×. (**a**) CC-0; (**b**) CC-3; (**c**) CC-6; (**d**) CC-12.

**Figure 6 materials-15-07255-f006:**
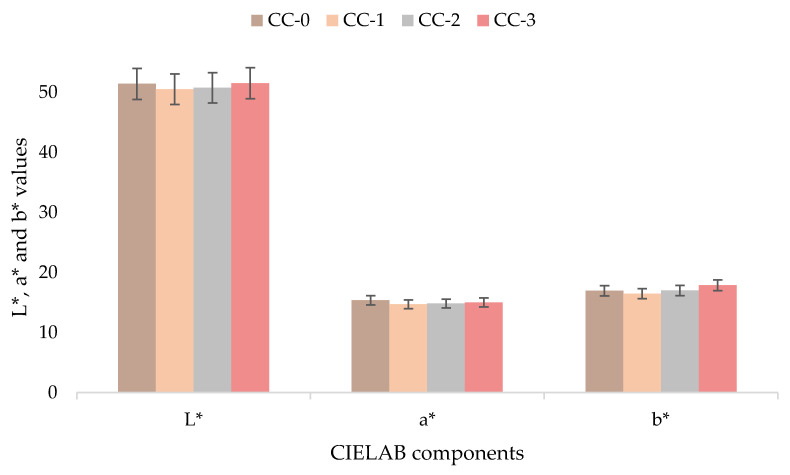
CIELAB component values for coated and uncoated leather samples according to the standard ISO 22700.

**Figure 7 materials-15-07255-f007:**
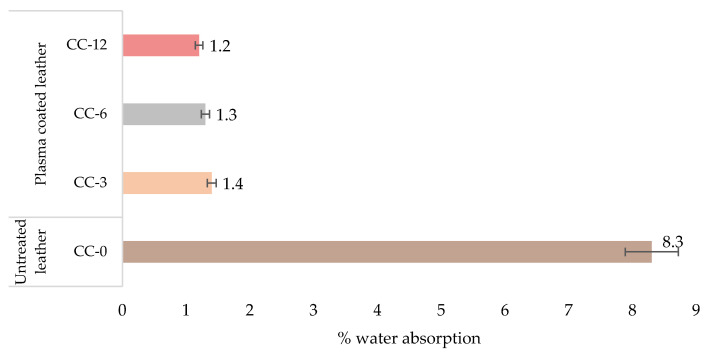
Water absorption (%) of plasma-coated and uncoated leather samples.

**Figure 8 materials-15-07255-f008:**
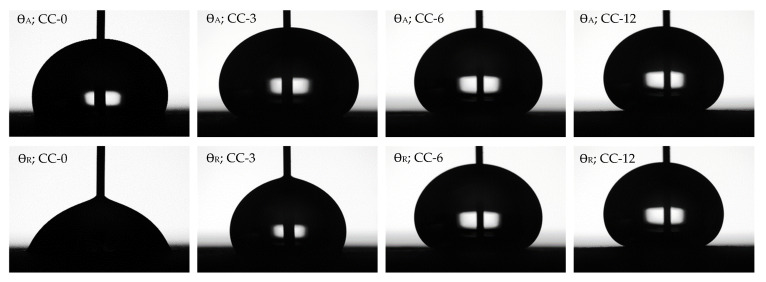
Advancing and receding drops of distilled water as standard liquid on uncoated and HMDSO plasma polymerised coated leather samples.

**Figure 9 materials-15-07255-f009:**
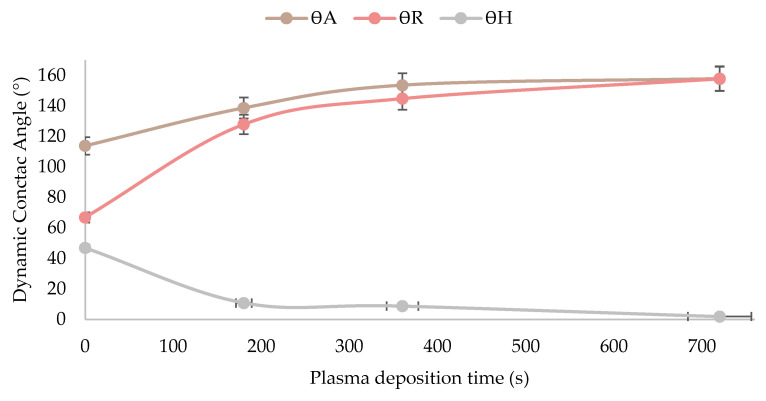
Evolution of the dynamic water contact angles as a function of HMDSO plasma deposition time on leather samples.

**Table 1 materials-15-07255-t001:** Samples nomenclature according to the operating conditions of the plasma coatings studied.

Leather Samples	Monomer	Time (s)	Power (W)
CC-0	-	-	-
CC-3	HMDSO	180	100
CC-6	HMDSO	360	100
CC-12	HMDSO	720	100

**Table 2 materials-15-07255-t002:** Elemental composition of untreated and plasma-coated leather samples.

Element	Leather Samples			
	CC-0	CC-3	CC-6	CC-12
C	75.52	84.09	74.20	70.44
O	20.41	8.94	13.26	13.63
N	3.12	0.00	0.00	0.00
Si	0.95	6.98	12.57	15.92
Si/C	0.01	0.08	0.17	0.23
Si/O	0.05	0.78	0.95	1.17

**Table 3 materials-15-07255-t003:** Atomic percentages (at.%) of XPS-identified chemical species at the C 1s and Si 2p peaks of untreated and plasma-coated leather samples.

Element	Species	Binding Energy (eV)	CC-0	CC-3	CC-6	CC-12
C 1s	C-H/C-C	284.6	55.17	44.44	71.05	67.79
	C-N/C-O/C-Si	286.0	11.23	11.66	0.66	1.48
	C=O	286.9	4.26	11.68	1.59	1.03
	O=C-O	288.7	4.86	10.07	0.65	-
	COOH/COOR	291.5	-	6.24	-	0.14
Si 2p	SiO_x_	101.3	-	6.98	12.57	15.92
	SiO_x_	102.2	0.95	-	-	-
	SiO_2_/SiO_2_(CH_3_)_3_	103.2	-	-	-	-

**Table 4 materials-15-07255-t004:** Water repellency rating values of uncoated and HMDSO plasma polymerised coated leather samples.

Water Repellency									
Sample/Solutions	0	1	2	3	4	5	6	7	8
CC-0	A	B	C	D					
CC-3	A	A	A	B	D				
CC-6	A	A	A	B	C	C	D		
CC-12	A	A	A	A	B	C	C	D	

**Table 5 materials-15-07255-t005:** Values of advance, receding and hysteresis contact angles of uncoated and plasma-coated leather samples.

Dynamic Water Contact Angle (DWCA)/(° Grades)	Leather Samples			
	CC-0	CC-3	CC-6	CC-12
ϴ_A_	114 ± 1	139 ± 3	154 ± 3	158 ± 2
ϴ_R_	67 ± 2	128 ± 4	145 ± 3	158 ± 4
ϴ_H_	47 ± 2	11 ± 1	9 ± 2	2 ± 0.3

## Data Availability

The data presented in this study are available upon request from the corresponding author.

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
