# Peer review of "Organosilicon-Based Plasma Nanocoating on Crust Leather for Water-Repellent Footwear"

_materials, 2022, doi:10.3390/ma15207255_

Round 1

Reviewer 1 Report

The manuscript presents surface analysis of leather samples before and after coating with a Si based material, aiming at obtaining surface hydrophobicity. The coating was obtained using plasma and HMDSO as precursor.

The text is clearly presented. Some minor aspects could be improved to further increase the quality of the manuscript.

The term "plasma coating treatment" is wrong. The process is "plasma coating"; the substrate is coated, not treated. All text should be thus corrected.

Figure 1 a is superflue. It should not apear.

Table 1. Presents physical and mechanical properties of the leather samples used. The information presented would be of interest in the case such parameters are measured after coating the substratrates, as well. As long as such a data set is not available, the information given in this table is not usable for the scope of presented work. The paper would gain a higher interest if such parameters are made available also for the coated samples.

Chapter 2.2 should contain information on the type of plasma used (RF, capacitively coupled).

Information on size of learther samples used should also be included.

Comments related to the advantages of using plasma for coating of leather should not mention reduced consumtion of water and energy, as this process uses vacuum pumps and water cooling which are not neglijible. As the main resouces for obtaining electricity are not "significant environmental benefits", as stated in the Concluision section.

Author Response

Please see the attachment for the response to your revisions.

Reviewer 2 Report

Comments and Suggestions

In this study, functional nanocoatings for waterproof footwear leather materials were investigated by chemical plasma polymerization by implanting and depositing the organosilicon compound HMDSO using a low pressure plasma system. To this end, the effect of monomers on the leather plasma deposition time was evaluated and the resulting plasma polymers were characterized by using different experimental techniques. The resulting polysiloxane polymers exhibited hydrophobic properties on leather. Furthermore, these chemical surface modifications created on the substrate can produce water repellent effects without altering the visual leather appearance and physical properties. Both plasma coating treatments and nanocoatings with developed water repellency properties can be considered as a more sustainable, automated and less polluting alternative to chemical conventional treatments that can be introduced into product finishing processes in the footwear industry. 

The leather samples deposited in the study were evaluated by Fourier Transform Infrared Spectroscopy (FTIR), X-ray Photoelectron Spectroscopy (XPS) and Scanning Electron Microscopy (SEM). Leather samples were tested by standard tests for color change, water resistance, surface wetting resistance and dynamic water contact angle (DWCA). 

In Figure 4(f): Why sample CC-6 has two SiO2 peaks with binding energies of 103.00 and 103.70 eV, which are absent in samples CC-3 and CC-12. What is the CC-6 specificity of the samples? Could the peak be fitted by without SiO2 by changing the FWMHs of the SiOx and SiO2(CH3)3? 

The O 1s peak should be analyzed because the binding energies of C 1s and Si 2p are affected. 

Line 318: “It was observed that in (b), (c), and (d) images, the surface is smoother and the leather fibres appear more flattened and not as loose as in sample (a).” But, Smooth sample surfaces do not show clearly. 

The authors propose the use of low-pressure plasma technology to alter the surface properties of leather to make it hydrophobic, thereby increasing leather usage. In addition, this waterproof coating treatment also helps reduce the environmental impact of the process and product, thereby contributing to the decarbonization of footwear manufacturing. They compared traditional waterproofing processes with wet and chemical processes, and plasma coatings have significant environmental benefits. Nonetheless, can the authors briefly describe decarbonization comparing this study with the traditional manufacturing process?

Author Response

(The authors gave the same response as above.)

Reviewer 3 Report

The manuscript presents an interesting study about organosilicon-based plasma nanocoating deposited on leather. The materials were characterized by FTIR, XPS, SEM, DWCA etc. The paper needs minor revisions before it is processed further, some comments follow:

Abstract:

The abstract must be improved. Please highlight the novelty of the study;

 Results and discussion

3.1. Compare the results obtained with others.

Figure 5. Introduce figure labels to highlight the interest zone for the reader.

Conclusions

The conclusion must be improved. Conclusions can be written with points. Also, add limitations and suggestions and quantitative results.

Author Response

(The authors gave the same response as above.)

Round 2

Reviewer 2 Report

accept

Author Response

Thank you for accepting comments.